# Prognostic Nomogram Combining Preoperative Neutrophil to Lymphocyte Ratio and Clinicopathologic Features for Gastric Cancer Patients after Distal Radical Gastrectomy: Based on Propensity Score Matching

**DOI:** 10.3390/jpm13010086

**Published:** 2022-12-29

**Authors:** Yi Liu, Chuandong Wang, Huan Wang, Changshun Yang, Xuefei Cheng, Weihua Li

**Affiliations:** 1Shengli Clinical Medical College of Fujian Medical University, Fuzhou 350000, China; 2Department of Gastrointestinal Surgery, People’s Hospital of Macheng City, Huanggang 438300, China; 3Department of Surgical Oncology, Fujian Provincial Hospital, Fuzhou 350000, China; 4Cardiac Center, Guangdong Women and Children’s Hospital, Guangzhou 510000, China

**Keywords:** gastric cancer (GC), preoperative inflammatory factors, neutrophil to lymphocyte ratio (NLR), prognosis, nomogram

## Abstract

**Background:** Preoperative inflammatory status has been widely used in assessing the prognosis of malignant tumor. This study aimed to establish a novel nomogram combining preoperative inflammatory factors and clinicopathologic features to predict the prognosis of gastric cancer (GC) patients after distal radical gastrectomy. **Methods:** A total of 522 GC patients from Fujian Provincial Hospital were retrospectively reviewed. Propensity score matching was performed and Cox regression models were used to analyze the clinical and pathological factors to determine their impact on survival. A prognostic nomogram was established and validated based on these factors. **Results:** The multivariate analysis indicated that tumor stage, pathological type, and neutrophil to lymphocyte ratio (NLR) were independent risk factors for the prognosis of GC patients. The nomogram was established based on these factors. In the primary cohort, the concordance index (C-index) of the nomogram was 0.753 (95% CI 0.647–0.840), which was higher than that of the American Joint Committee on Cancer (AJCC) tumor-node-metastasis (TNM) stage. The calibration curve showed the actual overall survival (OS) probabilities were in good keeping with those predicted by the nomogram. Furthermore, we divided the patients into two distinct risk groups for OS according to the nomogram points: low and high risk. The OS rates were significantly different among the subgroups (*p* ˂ 0.001). **Conclusions:** We proposed a novel nomogram combining preoperative NLR and clinicopathologic features that is economical, routinely available, and highly predictive of OS in GC patients after distal radical gastrectomy. Compared with the current AJCC TNM staging, this model was more accurate in prognostic prediction.

## 1. Introduction

Gastric cancer (GC) grows complex, like an incurable wound, leading to a systemic inflammatory immune response, which plays a major role in the progression of GC. Previous study has revealed that increased interaction between systemic inflammatory responses is correlated with poor outcomes in cancer patients [1]. Presently, the main comprehensive treatment of GCs is based on surgery, but surgery generates a mighty blow to the body, followed by causing a corresponding inflammatory response and stimulating the immune system to cause a metabolic stress response. Furthermore, relevant research indicated that poor nutritional status always led to poor prognosis of patients after critical surgery [2]. Furthermore, some studies have indicated that perioperative nutritional support may affect the long-term prognosis of cancer patients [3,4]. Thus, preoperative inflammation and nutritional status are very important to evaluate the therapeutic effect.

A variety of indicators have been used in GC as clinical predict prognostics such as preoperative neutrophil to lymphocyte ratio (NLR), platelet to lymphocyte ratio (PLR), prognostic nutrition index (PNI), albumin to globulin ratio (AGR), tumor-node-metastasis (TNM) stage, and so on [5,6,7,8,9,10,11,12,13,14,15]. However, previous study has shown that the TNM stage is not particularly sensitive for the prognosis of GC [7], which means that it is not enough to predict patient outcomes based on TNM classification alone. Moreover, due to the heterogeneity of the tumor, even patients with the same TNM stage or with the same treatment strategy may have different outcomes [16,17]. Some studies have indicated that high preoperative NLR and PLR are associated with a poor prognosis and high PLR is associated with progressive GC metastasis [5,18]. Lee et al. supposed that preoperative NLR and PLR were independent prognostic factors for overall survival (OS) in advanced GC [19]. Several studies have revealed that AGR and PNI are prognostic factors for various cancers including colorectal cancer, small-cell lung cancer, renal cell cancer, and glioblastoma [20,21,22,23]. PNI was also recognized as a valuable prognostic predictor for cancers of the digestive system [13,24]. A few studies have reported that low AGR is an independent prognostic factor for the assessment of cancers [25,26]. Recently, it has been shown that the nomogram, which is a simple graphical visualization combining and quantifying all independent prognostic factors, plays an increasingly vital role in medical sciences and clinical studies [27]. Although there are some prognostic nomograms available for gastric cancer [28,29,30], only a few of them considered the patient’s inflammatory and nutritional factors, which are simple and effective risk factors. Therefore, in order to predict the prognosis of GC accurately, we incorporated the clinicopathologic features and preoperative inflammatory and nutritional factors to propose a nomogram and perform tests to validate whether this model could predict the prognosis more accurately compared with traditional TNM staging systems.

## 2. Material and Methods

### 2.1. Patient Population

A retrospective study was performed in 522 patients with histologically diagnosed GC from 2014 to 2019 at the Fujian Provincial Hospital (Fuzhou, Fujian, China). Among them, 389 GC patients treated at the Department of Surgical Oncology between February 2014 and April 2019 formed the primary cohort, and 133 GC patients treated at the Department of Gastrointestinal Surgery between October 2014 and March 2017 formed the validation cohort. The inclusion criteria were as follows: (1) Patients who were diagnosed with GC histologically; (2) did not received any treatment preoperatively; (3) distal radical gastrectomy for GC (R0 resection + D2 lymph node dissection) [27]. The exclusion criteria were as follows: (1) Dependence on enteral nutrition (EN), parenteral nutrition (PN) or presence of acute inflammation (elevated serum C-reaction protein or procalcitonin) for 2 weeks; (2) patients diagnosed with a second tumor or an indefinite disease; (3) patients who received palliative resections; and (4) patients without complete clinical data. Written informed consent for participation or publication was provided by all patients. This study was reviewed and approved by the Ethics Committee of Fujian Provincial Hospital. All study procedures were performed in accordance with the 1964 Declaration of Helsinki and later versions.

### 2.2. Data Collection

The clinical characteristics included age, gender, tumor size, Borrmann type, TNM stage, pathological type, histological grade, and OS status. The levels of preoperative inflammatory factors were collected. The results of preoperative blood tests including the serum albumin level and serum globulin level, neutrophil, lymphocyte, and platelet blood cell count was obtained within 1 week before surgery. NLR is the neutrophil count (N)/lymphocyte count (L). PLR is the platelet count (PLT)/lymphocyte count (L). PNI is the Onodera Prognostic Nutrition Index (PNI), PNI = 5*Lymphocyte count (L × 109/L) + serum albumin (ALB g/L). AGR is serum albumin (ALB g/L)/serum globulin (GLB g/L).

### 2.3. Follow-Up

After completion of primary treatment, all patients with GC were followed up periodically according to the clinical guidelines. During the first 2 years, the patients were followed up every 3 months. Patients with no recurrence during the next 3–5 years were generally followed up every 6 months and annually thereafter. Patients who did not attend our hospital on time were followed-up by telephone to obtain the information about their treatment and survival status. The duration of the follow-up in our study was measured as the overall survival (OS), taking into account the time from GC diagnosis to last follow-up or death.

### 2.4. Statistical Analysis

Statistical analyses were performed using SPSS 26.0 (IBM, Chicago, IL, USA) and R for Windows (version 4.2.0). The propensity score was calculated based on age, gender, tumor size, tumor stage, node stage, metastasis stage, clinical stage, Borrmann type, pathological type, histological grade, OS status, NLR, PLR, PNI, and AGR. Patients in the primary cohort were matched 1:1 using nearest neighbor matching, based on the closest propensity score to those in the validation cohort. X-tile statistical software (version 3.6.1) was used to evaluate the optimal cutoff points of this study, and continuous variables were converted into categorical variables, which were classified according to the clinical results. We used univariate and multivariate regression analyses to analyze the risk factors in the primary cohort. A nomogram was established based on the results of multivariate analysis by the rms package. Discrimination and calibration tests were used to validate the accuracy of the nomogram in the primary as well as external validation cohorts. We used Harrell’s concordance index (C-index) to measure the discrimination of the nomogram. The value of the C-index ranged from 0.5 to 1.0, where 0.5 indicates random chance, while 1.0 means that the model was fully capable of predicting the outcome correctly. The calibration curve of the nomogram for predicting the OS was drawn. Then, we calculated the total points of each patient based on the established nomogram model, and used the X-tile program to delineate two groups of patients with different prognostic risks based on the total points. The Kaplan–Meier method and log-rank test were used to compare the survival curves of the dichotomous risk groups. All statistical tests were two-sided, and *p* < 0.05 indicates a statistically significant difference.

## 3. Results

### 3.1. Basic Clinical Characteristics

The primary cohort included 389 GC patients treated at the Fujian Provincial Hospital, Department of Surgical Oncology between February 2014 and April 2019. The validation cohort included 133 GC patients treated at the Fujian Provincial Hospital, Department of Gastrointestinal Surgery between October 2014 and March 2017. Before the propensity score matching, the age, gender, tumor size, node stage, metastasis stage, clinical stage, pathological type, OS status, NLR, PNI, and AGR were not significantly different between both groups (Table 1). Compared to the primary cohort, the tumor stage of patients in the validation cohort was significantly earlier (*p* = 0.046). The Borrmann types of patients in the validation cohort were significantly heavier (*p* = 0.034) and the histological grades of the patients were significantly better in the validation cohort (*p* < 0.001). The PLR of patients in the primary cohort were significantly higher than those in validation cohort (*p* = 0.047). A propensity score was calculated to adjust for biases caused by differences in the baseline characteristics between the two cohorts. After matching, there were no significant differences in the baseline characteristics between both groups (Table 1).

### 3.2. Biomarker Selection

Clinicopathologic characteristics, inflammatory factors, and nutritional factors were used to perform univariate and multivariate Cox proportional hazard regression analyses (Table 2). Univariate analysis showed significant correlation among tumor stage, node stage, pathological type, NLR, PLR, PNI, and AGR (*p* ˂ 0.05). The variables distinguished in the univariate analyses were evaluated by multivariate analysis. The results indicated that tumor stage, pathological type, and NLR were independent risk factors for the prognosis of GC patients.

### 3.3. Development of the Prediction Model

To predict the probability of OS in the GC patients, we included the variables of tumor stage, pathological type, and NLR in our nomogram. A backward step-down selection process with the Akaike information criterion (AIC) was used to perform the construction of the final nomogram model. Finally, we constructed a nomogram to predict the 1-, 3-, and 5-year OS (Figure 1).

### 3.4. Internal and External Validation of the Nomogram Model

We used Harrell’s concordance index to compare the accuracies of the OS predictions in GC patients between our nomogram and existing TNM stage (Table 3). In the primary cohort, the nomogram had a C-index of 0.753 (95% CI 0.647–0.840), which was higher than that of the TNM stage (0.689, 95% CI 0.600–0.771). In the validation cohort, the C-index of the nomogram model (0.748, 95% CI 0.649–0.838) was also greater than that of the TNM stage (0.727, 95% CI 0.650–0.803). Calibration curves described the calibration of our model based on the agreement between the predicted OS and the observed survival outcomes, where the x-axis represents the actual survival while the y-axis represents the observed survival. The gray and blue lines were close in both the primary and validation cohorts, which means that the calibration plot for the probability of OS at 1, 3, or 5 years after therapy showed the optimal agreement between the actual observation and nomogram prediction (Figure 2 and Figure 3).

### 3.5. Risk Stratification of OS by the Nomogram Model

According to the nomogram established in this study, we divided the patients into low- and high-risk groups. The low-risk group had the longest OS compared with the high-risk group in both the primary cohort and validation cohort (63.026 ± 1.816 months vs. 39.674 ± 3.811 months and 56.904 ± 1.151 months vs. 43.647 ± 3.115 months) (Table 4). Moreover, the Kaplan–Meier curves were plotted and there were significant differences among these two groups (*p* ˂ 0.001) (Figure 4).

## 4. Discussion

Accurate prediction of the prognosis of cancer patients is of great significance to determine the definitive treatment or management plan [31,32]. Although most of us used traditional TNM staging systems to predict the prognosis of GC patients clinically, several studies have recently supposed that some drawbacks may exist when predicting the prognosis of GC patients only according to the TNM classification [17,18]. Previous studies have indicated that preoperative immunological and nutritional conditions are associated with both the postoperative and long-term outcomes of malignant tumors [33,34,35,36]. In our study, we found that NLR, PNI, and AGR were associated with the OS of GC patients in univariate analysis, but only NLR was an independent prognostic factor in multivariate analysis, together with tumor stage and pathological type. As we know, a nomogram can be established according to a Cox regression model, which is considered as a prediction model in clinic. The nomogram consists of coordinate axes and a scoring system. Each axis represents an independent survival predictor, and the corresponding score on the axis represents the impact of the predictor. As a result, we can see a perceptible visualization of the survival of a specific disease easily by nomogram. Thus, we established a nomogram according to the weights of these factors in the model, which was simple and effective to evaluate. All factors included in the model are easily available in clinical practice, and the internal validation showed consistent and stable predictive power, making it a practical tool for clinical reference.

According to our results, this model performed well in predicting the OS of GC patients, with a C-index of 0.753 in the primary cohort and 0.748 in the validation cohort. Moreover, our model predicted the OS of GC patients more accurately than the TNM stage (0.689 in primary cohort and 0.727 in validation cohort). Furthermore, we found that the nomogram prediction and actual observation were fairly close according to the calibration curves. The TNM stage only reflects the depth of tumor invasion, lymph node metastasis, and whether there is distant metastasis. Mao YP et al. observed that patients with the same TNM stage may have different clinical outcomes; conversely, patients with the same clinical outcome can be classified into different TNM stages [17]. Therefore, the traditional TNM staging system sometimes fails to accurately predict the OS of GC patients, and the model in this study can remedy this deficiency.

Actually, previous studies have developed several other nomograms to predict the individual survival of patients with gastric cancer. However, there have been some shortcomings. In a study published in 2021, the model was validated internally using the bootstrap method, lacking external validation in an independent cohort from a different institution [29]. Another study developed and validated a nomogram for GC patients using a multicenter database in Korea [30], however, it mainly focused on the clinical features and pathological results. Furthermore, this study only included patients who underwent open gastrectomy. It is worth noting that although a study published in 2022 was to use a web-based nomogram so that any expert could calculate the overall survival probability and had a long-term follow-up period, it did not include some clinical and pathological characteristics into the model [28]. In contrast, we incorporated clinicopathologic features and preoperative inflammatory factors to propose a nomogram, which could predict the prognosis of gastric cancer patients after distal radical gastrectomy. The results of our study indicated that patients with GC had different prognosis of OS by varying perioperative inflammatory and nutritional status (Table 2). Furthermore, due to combining the clinical inflammatory factors, the pathological type of GC, and traditional TNM stage, our method took into account the anatomical and individual patient differences to predict the 1-, 3-, and 5-year OS of GC patients more accurately. It is worth noting that the differences between the groups predicted by this nomogram for different prognoses were significant in the primary cohort and the validation cohort (Table 4), which means that this model performed well in predicting the overall survival. Thus, we supposed that patients with a high number of total points according to this model were classified as high risk and should be given active therapy and special attention.

Several studies have reported that chronic inflammation is closely associated with GC invasion and metastasis [37,38]. Inflammatory cells such as neutrophils, lymphocytes, and platelets continue to produce a range of cytokines and chemokines, which promote tumor growth, invasion, and metastasis when the tumor microenvironment is forming [39]. Tumor growth could easily lead to impaired nutrient absorption and a broken immuno-nutritional status, which would lead to a persistent chronic inflammatory response and promote tumor growth. The inflammation caused by surgery corresponds to the degree of surgical trauma, leading to metabolic stress in the body. In order to heal the trauma caused by surgery, the body needs to carry out metabolism, so that the body can use protein and muscle to restore the normal nutritional state of the body in a short period of time. There is increasing evidence that the prognosis of tumor patients is closely related to their nutritional status [34,35,36]. However, it is still unknown as to why preoperative malnutrition leads to poor postoperative outcome. Previous study has suggested that malnutrition impaired immune function, which led to an increased risk of postoperative infection and tumor metastasis [40]. Some studies have considered that malnutrition is a chronic or subacute state. Although the degree of its inflammatory response is different, it could lead to changes in components of the human body and the decline in immune function [41,42]. Meanwhile, malnutrition could activate the systemic inflammatory response and influence host immunity [14]. In terms of the systemic inflammatory and nutritional factors, our univariate analysis showed that NLR, PNI, and AGR were associated with the prognosis of GC patients, but only NLR was an independent prognostic factor after multivariate analysis. Previous studies considered NLR as a highly reproducible, cost-effective, and widely available prognostic marker for GC patients [43,44,45]. Relevant research has also indicated that the preoperative NLR correlated with not only the long-term outcomes, but also the perioperative outcomes of GC patients [46].

The mechanisms by which systemic inflammatory responses are associated with tumor progression have been extensively discussed. One reason for this correlation is that in patients with high NLR, tumor growth may be supported by neutrophil-derived cytokines such as vascular endothelial growth factor, interleukin-18, and matrix metalloproteinases [46]. Additionally, the increased number of neutrophils around the tumor may suppress the anti-tumor immune responses of natural killer cells and activated T cells [46]. At the same time, the decrease in the number of lymphocytes may weaken the lymphocyte-mediated antitumor cellular immune response. Therefore, it may be that neutrophilia and lymphopenia work together to increase NLR, thereby promoting angiogenesis, suppressing antitumor responses, and ultimately promoting tumor growth and progression [43,44,45].

Despite several advantages of our model, there were still some limitations to the current study. First, this study was a retrospective analysis, and there may have been sample selection bias in our study. Second, both cohorts in this study were from the same medical center, and the sample sizes of the two cohorts were not large enough. Thus, for this model to be valid, more data from other institutions in other regions will be needed in the future.

## 5. Conclusions

In conclusion, we established a novel prognostic nomogram to predict the prognosis of patients with GC after distal radical gastrectomy. Compared with the current TNM staging system, this nomogram is more effective in predicting prognosis. As an economical, simple, and routinely available prognostic tool, it has certain clinical application in improving the prediction of prognosis and guiding treatment strategies.

## Figures and Tables

**Figure 1 jpm-13-00086-f001:**
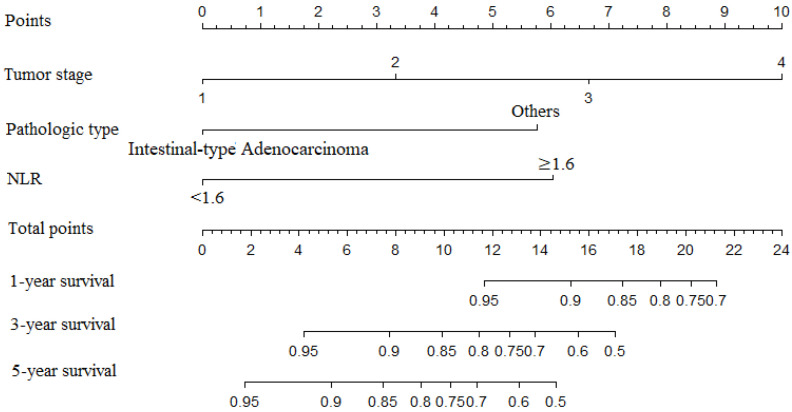
Nomogram included tumor stage, pathological type, and neutrophil to lymphocyte ratio (NLR) by the primary cohort, which were used for the prediction of the 1-, 3-, and 5-year overall survival (OS) in the GC patients.

**Figure 2 jpm-13-00086-f002:**
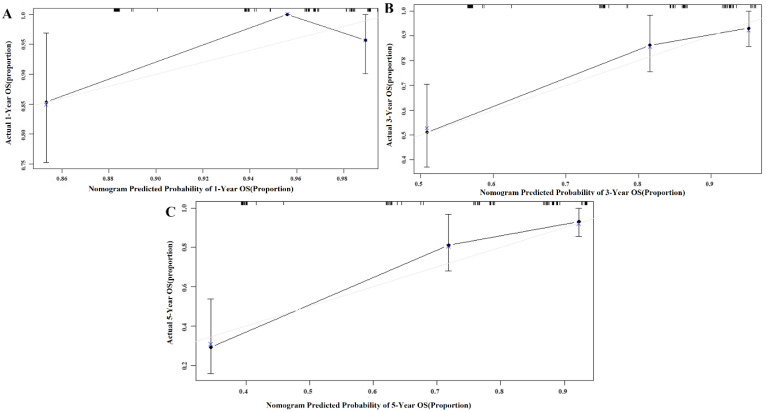
Calibration curve of the nomogram in the primary cohort, with the *x*-axis representing the actual survival estimated by the nomogram and the *y*-axis representing the observed survival calculated by the Kaplan–Meier method. (**A**) 1-year OS in the primary cohort. (**B**) 3-year survival OS in the primary cohort. (**C**) 5-year survival OS in the primary cohort.

**Figure 3 jpm-13-00086-f003:**
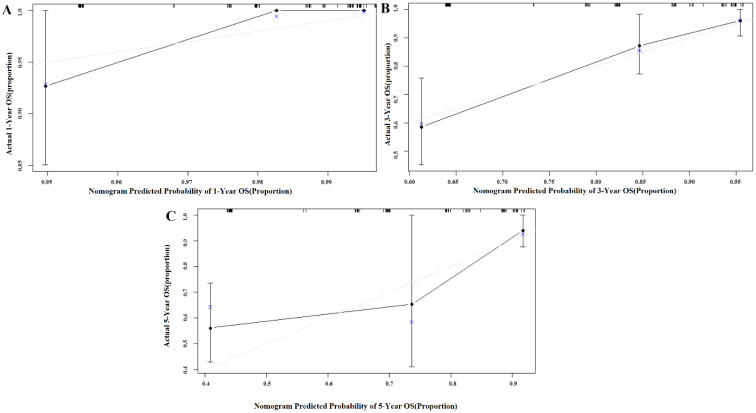
Calibration curve of the nomogram in the validation cohort, with the *x*-axis representing the actual survival estimated by the nomogram and the *y*-axis representing the observed survival calculated by the Kaplan-Meier method. (**A**) 1-year OS in the validation cohort. (**B**) 3-year survival OS in the validation cohort. (**C**) 5-year survival OS in the validation cohort.

**Figure 4 jpm-13-00086-f004:**
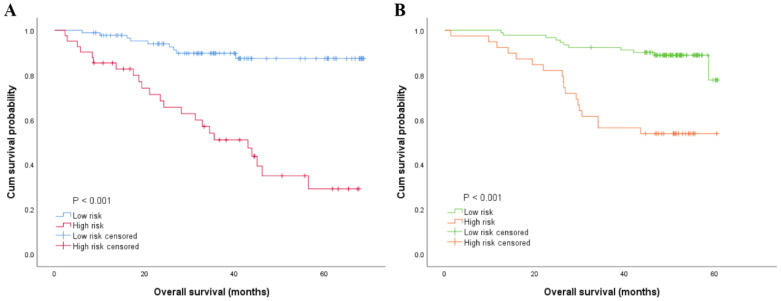
Kaplan–Meier curve was used to evaluate the prognosis based on the predictors from the nomogram model in the primary cohort (**A**) and those in the validation cohort (**B**).

**Table 1 jpm-13-00086-t001:** The clinicopathological characteristics of the patients.

	Raw Data			After Propensity-Matching		
Characteristics	Primary Cohort	Validation Cohort	*p* Value	Primary Cohort	Validation Cohort	*p* Value
No. of patients	389	133		130	130	
Age (years)			0.970			0.535
≥60	204	70		66	69	
˂60	185	63		64	61	
Gender			0.759			0.075
Male	263	88		99	86	
Female	126	45		31	44	
Tumor size (cm)			0.503			0.615
≥3	238	77		78	74	
˂3	151	56		52	56	
Tumor stage			0.046			0.293
T1	109	38		33	38	
T2	65	25		24	25	
T3	82	14		25	14	
T4	133	56		48	53	
Node stage			0.489			0.826
N0	190	65		63	63	
N1	58	20		21	20	
N2	61	27		22	27	
N3	80	21		24	20	
Metastasis stage			0.445			1.000
M0	388	132		130	129	
M1	1	1		0	1	
Clinical stage			0.436			0.588
Ⅰ	144	47		48	47	
Ⅱ	92	38		30	36	
Ⅲ	152	47		52	46	
Ⅳ	1	1		0	1	
Borrmann type			0.034			0.457
Ⅰ	38	15		10	15	
Ⅱ	108	20		28	20	
Ⅲ	189	76		71	76	
Ⅳ	54	22		21	19	
* Pathological type			0.470			0.586
Intestinal-type adenocarcinoma	323	114		114	111	
Others	66	19		16	19	
Histological grade			<0.001			0.791
Well	193	92		87	89	
Moderately-poor	196	41		43	41	
OS status			0.865			0.661
Survived	304	103		98	101	
Dead	85	30		32	29	
NLR			0.085			0.653
Median (P_25_, P_75_)	1.93 (1.43, 2.61)	1.78 (1.28, 2.58)		1.80 (1.32, 2.33)	1.78 (1.27, 2.60)	
PLR			0.047			0.992
Median (P_25_, P_75_)	132.50 (101.00, 178.61)	121.74 (93.55, 154.75)		122.47 (91.88, 166.20)	121.00 (93.29, 156.25)	
PNI			0.250			0.970
Median (P_25_, P_75_)	52.00 (47.00, 55.50)	53.00 (48.00, 56.50)		53.00 (47.38, 57.00)	53.00 (48.00, 56.50)	
AGR			0.916			0.636
Median (P_25_, P_75_)	1.64 (1.43, 1.77)	1.62 (1.46, 1.76)		1.64 (1.41, 1.77)	1.62 (1.46, 1.76)	

Notes: * Pathological type, according to the Lauren classification. Abbreviations: NLR: Neutrophil to lymphocyte ratio; PLR: Platelet to lymphocyte ratio; PNI: Prognostic nutrition index; AGR: Albumin to globulin ratio.

**Table 2 jpm-13-00086-t002:** Univariate and multivariate Cox hazard analysis of the primary cohort.

Characteristics	Univariate Analysis		Multivariate Analysis	
	HR (95% CI)	*p* Value	HR (95% CI)	*p* Value
No. of patients				
Age (years)				
˂60	1			
≥60	1.754 (0.864–3.561)	0.120		
Gender				
Male	1			
Female	0.833 (0.385–1.801)	0.642		
Tumor size (cm)				
˂3	1			
≥3	1.751 (0.810–3.787)	0.154		
Tumor stage				
T1	1		1	
T2	0.696 (0.116–4.188)	0.693	0.616 (0.101–3.771)	0.601
T3	1.471 (0.297–7.295)	0.637	1.076 (0.216–5.356)	0.929
T4	5.186 (1.558–17.270)	0.007	4.084 (1.188–14.041)	0.026
Node stage				
N0	1			
N1	2.295 (0.700–7.523)	0.170		
N2	6.592 (2.430–17.880)	˂0.001		
N3	4.783 (1.738–13.163)	0.002		
Borrmann type				
Ⅰ	1			
Ⅱ	0.643 (0.153–2.701)	0.547		
Ⅲ	0.793 (0.232–2.708)	0.711		
Ⅳ	1.193 (0.308–4.619)	0.798		
* Pathological type				
Intestinal-type Adenocarcinoma	1		1	
Others	2.872 (1.235–6.680)	0.014	3.398 (1.427–8.089)	0.006
Histological grade				
Well	1			
Moderately-poor	1.319 (0.644–2.702)	0.450		
NLR				
˂1.6	1		1	
≥1.6	4.673 (1.794–12.168)	0.002	3.396 (1.283–8.988)	0.014
PLR				
˂92.9	1			
≥92.9	2.505 (0.961–6.528)	0.060		
PNI				
≥48	1			
˂48	2.307 (1.177–4.641)	0.019		
AGR				
≥1.4	1			
˂1.4	2.169 (1.060–4.439)	0.034		

Notes: * Pathological type, according to the Lauren classification. Abbreviations: NLR: Neutrophil to lymphocyte ratio; PLR: Platelet to lymphocyte ratio; PNI: Prognostic nutrition index; AGR: Albumin to globulin ratio; HR: hazard ratio; CI: confidence interval.

**Table 3 jpm-13-00086-t003:** C-index of the nomogram and TNM stage in the primary cohort and validation cohort.

	Primary Cohort		Validation Cohort	
	C-Index	95% CI	C-Index	95% CI
Nomogram	0.753	0.642–0.833	0.748	0.638–0.827
TNM stage	0.689	0.583–0.766	0.727	0.632–0.788

**Table 4 jpm-13-00086-t004:** Overall survival time (OS) in the primary cohort and validation cohort.

Groups		OS (Mean ± SD)	1-Year (%)	3-Year (%)	5-Year (%)
Primary cohort	Low-Risk	63.026 ± 1.816	97.7	89.7	87.4
	High-Risk	39.674 ± 3.811	85.4	51.0	29.2
Validation cohort	Low-Risk	56.904 ± 1.151	98.9	92.3	77.7
	High-Risk	43.647 ± 3.115	92.3	56.4	53.8

## Data Availability

Please contact the corresponding author (Weihua Li, email: liwh@fjmu.edu.cn) for data requests.

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
