# Peer review of "Prognostic Nomogram Combining Preoperative Neutrophil to Lymphocyte Ratio and Clinicopathologic Features for Gastric Cancer Patients after Distal Radical Gastrectomy: Based on Propensity Score Matching"

_jpm, 2022, doi:10.3390/jpm13010086_

Round 1
Reviewer 1 Report
The authors created a nomogram to predict the overall survival of newly diagnosed gastric carcinoma patients by considering immunological, nutritional, and clinicopathologic data. The idea is of clinical importance, and altogether, the methods sound appropriate and adequate. Nevertheless, there are already several similar nomograms on record, and some major points remain to be addressed especially regarding the tumor type grouping.
Comments:
1. It is suggested not to use abbreviations in the title. Please include the full form for NLR there.
2. Several other prognostic nomograms are available for gastric cancer. It is recommended to discuss the advantages and any probable shortcomings of the present nomogram compared to them.
3. According to inclusion criteria 3, the patients had undergone “distal radical gastrectomy”, and logically they were non-cardia tumors. If true, it is suggested to add “distal” in the title. On the other hand, if “distal” is written by mistake and as a result total gasterectomies are also mentioned, tumors in the cardia are probably included in the study. Considering the difference in the biology and clinical behavior of cardia tumors compared to their non-cardia counterparts, it is highly recommended to include the site of the tumor (cardia vs non-cardia) in the analysis.
4. Signet-ring cell/diffuse-type carcinoma is indeed a type of poorly-differentiated adenocarcinoma. Therefore, it is not accurate to divide gastric carcinomas into “Adenocarcinoma” and “Others”, while signet-ring cell carcinomas are included in the Others group. The groups could have been “Intestinal-type Adenocarcinoma” vs. “Others” considering the Lauren classification.
5. Intramucosal carcinomas are T1a adenocarcinoma tumors (tumors that invade the lamina propria but are limited to mucosal layer). Therefore, they should be included in the “Adenocarcinoma” group as well.
6. Table 2 shows a worse hazard ratio for “Others” patients compared to “Adenocarcinoma” patients in both uni- and multivariate analysis. The “Others” group is a mixture of different types of tumors including the signet-ring cell/diffuse-type carcinoma, neuroendocrine carcinoma, and adenosquamous carcinoma, which are all known to harbor a worse prognosis than intestinal-type adenocarcinomas. . I guess it might be the inclusion of such ominous tumors in the “Others” group that makes tumor type so much more significant. I suggest performing a subset analysis on the “Others” group. Another suggestion would be not to include uncommon tumor types in the study
7. I wonder whether the primary and validation cohorts were matched for the follow-up time.
8. The authors highlight that the present model is superior to the current AJCC TNM staging in terms of prognostic value. However, the differences in concordance indices look modest, according to Table 3.
9. Line 275: I think “neutropenia” should be replaced by “neutrophilia”.
Author Response
The authors created a nomogram to predict the overall survival of newly diagnosed gastric carcinoma patients by considering immunological, nutritional, and clinicopathologic data. The idea is of clinical importance, and altogether, the methods sound appropriate and adequate. Nevertheless, there are already several similar nomograms on record, and some major points remain to be addressed especially regarding the tumor type grouping.
Response: Thank you for your comments for our manuscript. Those comments are all valuable and very helpful for revising and improving our paper, as well as the important guiding significance to our researches. We have studied comments carefully and made correction which we hope meet with approval. The main corrections in the paper and responds to the comments are as following:
Comments:
- It is suggested not to use abbreviations in the title. Please include the full form for NLR there.
Response: Thank you for your comments on this study. We have made revisions in title.
- Several other prognostic nomograms are available for gastric cancer. It is recommended to discuss the advantages and any probable shortcomings of the present nomogram compared to them.
Response: Thank you for your comments on this study. Your comments were highly insightful and enabled us to greatly improve the quality of our manuscript. Although several other nomograms have been previously developed to predict the individual survival of patients with gastric cancer, they have their own advantages and disadvantages. Some focus on the relationship between clinical features and prognosis, some on pathological results, and still others on different treatment modalities. In contrast, we incorporated clinicopathologic features and preoperative inflammatory factors to propose a nomogram and perform some tests to validate whether this model can predict prognosis of gastric cancer patients after distal radical gastrectomy more accurately compared with traditional TNM staging systems. We have made revisions in the discussion section in manuscript. Thank you very much!
- According to inclusion criteria 3, the patients had undergone “distal radical gastrectomy”, and logically they were non-cardia tumors. If true, it is suggested to add “distal” in the title. On the other hand, if “distal” is written by mistake and as a result total gasterectomies are also mentioned, tumors in the cardia are probably included in the study. Considering the difference in the biology and clinical behavior of cardia tumors compared to their non-cardia counterparts, it is highly recommended to include the site of the tumor (cardia vs non-cardia) in the analysis.
Response: Thank you for your comments on this study. We really apologize for our errors in expression. The study included patients who had undergone distal radical gastrectomy. We corrected mistakes and made revisions in manuscript.
- Signet-ring cell/diffuse-type carcinoma is indeed a type of poorly-differentiated adenocarcinoma. Therefore, it is not accurate to divide gastric carcinomas into “Adenocarcinoma” and “Others”, while signet-ring cell carcinomas are included in the Others group. The groups could have been “Intestinal-type Adenocarcinoma” vs. “Others” considering the Lauren classification.
Response: Thank you for your comments on this study. We quite agree with your suggestion. We have re-checked the case data, and the pathological types of the cases in the "others" group are mainly signet-ring cell/diffuse-type carcinoma. According to the Lauren classification, we divided these cases into “Intestinal-type Adenocarcinoma” vs. “Others”. Thank you very much for your advice. We have made revisions in manuscript.
- Intramucosal carcinomas are T1a adenocarcinoma tumors (tumors that invade the lamina propria but are limited to mucosal layer). Therefore, they should be included in the “Adenocarcinoma” group as well.
Response: Thank you for your comments on this study. We are really sorry for such mistakes. This is not “intramucosal carcinomas” here. Originally, we wanted to express “mucinous adenocarcinoma”, and this was a spelling mistake. We really apologize for it. Finally, we have re-checked the cases and divided these cases into “Intestinal-type Adenocarcinoma” vs. “Others” according to Lauren classification.
- Table 2 shows a worse hazard ratio for “Others” patients compared to “Adenocarcinoma” patients in both uni- and multivariate analysis. The “Others” group is a mixture of different types of tumors including the signet-ring cell/diffuse-type carcinoma, neuroendocrine carcinoma, and adenosquamous carcinoma, which are all known to harbor a worse prognosis than intestinal-type adenocarcinomas. I guess it might be the inclusion of such ominous tumors in the “Others” group that makes tumor type so much more significant. I suggest performing a subset analysis on the “Others” group. Another suggestion would be not to include uncommon tumor types in the study.
Response: Thank you for your comments on this study. After receiving your comments and suggestions, we have re-checked the cases included in this study. Actually, the pathological types of the cases in the "others" group are mainly signet-ring cell/diffuse-type carcinoma. Really thanks for your advice, we decided to divide these cases into “Intestinal-type Adenocarcinoma” vs. “Others” according to the Lauren classification. We have made revisions in manuscript. Your comments are really valuable and very helpful for revising and improving our paper. Thank you very much!
- I wonder whether the primary and validation cohorts were matched for the follow-up time.
Response: Thank you for your comments on this study. Yes, they were matched for the follow-up time.
- The authors highlight that the present model is superior to the current AJCC TNM staging in terms of prognostic value. However, the differences in concordance indices look modest, according to Table 3.
Response: Thank you for your comments on this study. We also acknowledged that although our results show that this nomogram was superior to the current AJCC TNM staging in terms of prognostic value, the difference was not significant enough. However, as far as the results of the current study are concerned, we think they still have certain clinical significance. While single-center studies do not permit a very strong level of evidence, we believe that they become a good starting point to conduct a well-designed multi-center studies with large sample on the topic and get a strong level of evidence in future.
- Line 275: I think “neutropenia” should be replaced by “neutrophilia”.
Response: Thank you for your comments on this study. We have made revision in manuscript. We are really appreciated for your help.
Reviewer 2 Report
The authors of this study try to present the evidence regarding the development of normogram to predict the prognosis of patients with GC. I think the topic of this manuscript is interesting. The Methodological quality is good enough and seems replicable with detailed presentation of the Results. The Discussion is also already comprehensive enough. I only have several minor comments regarding this manuscript.
1) Introduction: I think the background provided by the authors is not strong enough to justify this study. Why do we need the normogram from this study to predict the prognosis of GC? What is lacking from the previous studies? Is there any literature gap?
2) Methods: "...presence of acute inflammation for 2 weeks." How do the authors detect the presence of acute inflammation in the participants?
3) Discussion: In the Discussion, the authors may also discuss the results from previously published studies which also analyze the same topic about normogram to predict GC outcomes (https://doi.org/10.1371/journal.pone.0119671, https://doi.org/10.1038/s41598-022-08465-w, https://doi.org/10.1155%2F2021%2F2923700). The authors may discuss what made their study differ from those previously published studies.
Author Response
The authors of this study try to present the evidence regarding the development of nomogram to predict the prognosis of patients with GC. I think the topic of this manuscript is interesting. The Methodological quality is good enough and seems replicable with detailed presentation of the Results. The Discussion is also already comprehensive enough. I only have several minor comments regarding this manuscript.
Response: We are grateful to you for your valuable comments and suggestions which help us to improve the quality of the manuscript. We have studied the comments carefully and have made modifications and corrections which we hope meet your approval. We revised the manuscript according to your kind advices and detailed suggestions. The main corrections in the paper and responds to the comments are as following:
1) Introduction: I think the background provided by the authors is not strong enough to justify this study. Why do we need the nomogram from this study to predict the prognosis of GC? What is lacking from the previous studies? Is there any literature gap?
Response: Thank you for your comments on this study. Your suggestion really helps us to improve the quality of the manuscript. GC is always accompanied by changes in the immune system, such as the development of various inflammatory factors and the activation of immune cells. Furthermore, it has been generally acknowledged that adequately predicting the patient prognosis only according to the TNM classification is not enough. Recently, nomogram plays an increasingly vital role in medical sciences and clinical studies. Although there were some prognostic nomograms available for gastric cancer, few of them considered the patient’s inflammatory and nutritional factors, which were simple and effective risk factors. Therefore, in order to predict prognosis of GC accurately, we design this study. We have made revision in the “Introduction” part of manuscript.
2) Methods: "...presence of acute inflammation for 2 weeks." How do the authors detect the presence of acute inflammation in the participants?
Response: Thank you for your comments on this study. We are very sorry for the misunderstanding caused by our unclear expression. The presence of acute inflammation in the participants was detected by the detection of elevated serum C-reaction protein or procalcitonin. We have made revision in the “Methods” part of manuscript.
3) Discussion: In the Discussion, the authors may also discuss the results from previously published studies which also analyze the same topic about normogram to predict GC outcomes (https://doi.org/10.1371/journal.pone.0119671, https://doi.org/10.1038/s41598-022-08465-w, https://doi.org/10.1155%2F2021%2F2923700). The authors may discuss what made their study differ from those previously published studies.
Response: Thank you for your comments on this study. Your comments were highly insightful and enabled us to greatly improve the quality of our manuscript. Although several other nomograms have been previously developed to predict the individual survival of patients with gastric cancer, they have their own advantages and disadvantages. Some of them focused on the relationship between clinical features and prognosis, while others focused on pathological results or different treatment modalities. In contrast, we incorporated clinicopathologic features and preoperative inflammatory factors to propose a nomogram and perform some tests to validate whether this model can predict prognosis of gastric cancer patients after distal radical gastrectomy more accurately compared with traditional TNM staging systems. We have made revisions in the discussion section in manuscript. Thank you very much!
Round 2
Reviewer 1 Report
Thank you for the revisions. I think the manuscript is improved.